# Collaborative Learning for Semi-Supervised LiDAR Semantic Segmentation

**Bin Yang** [1] [2]   **Alexandru Paul Condurache** [1] [2]

## Abstract

Annotating large-scale LiDAR point clouds for 3D semantic segmentation is costly and time-consuming, which motivates the use of semi-supervised learning (SemiSL). Standard LiDAR SemiSL methods typically adopt a two-step training paradigm, where pseudo-labels are separately generated from a single distillation source, either from the same or another LiDAR representation. Such supervision relies on a unique source of pseudo-labels, which can reinforce confirmation bias and propagate errors during training, ultimately limiting performance. To address this challenge, we introduce *CoLLiS*, a novel framework that leverages **Co**llaborative **L**earning for **Li**DAR **S**emi-supervised segmentation. Unlike prior paradigms with decoupled pseudo-labeling and training phases, *CoLLiS* trains multiple representations collaboratively in a single step by treating them as coequal students. Each student is adaptively distilled from multiple representations, while inter-student disparities are monitored online to resolve contradictory supervision and effectively mitigate confirmation bias. Extensive experiments on three datasets demonstrate that *CoLLiS* consistently outperforms state-of-the-art LiDAR SemiSL methods, with particularly strong gains in low-label regimes.

## 1. Introduction

The robustness of LiDAR sensors in environmental perception has propelled their widespread adoption for 3D scene understanding in autonomous driving. The inherent geometric challenges of LiDAR data, such as sparsity and viewpoint distortions, have motivated extensive research into diverse input representations for semantic segmenta-

[1]Bosch Research, Robert Bosch GmbH, Stuttgart, Germany [2]Institute for Neuro- and Bioinformatics, University of Lübeck, Lübeck, Germany. Correspondence to: Bin Yang <bin.yang3@de.bosch.com>.

*Proceedings of the $43^{rd}$ International Conference on Machine Learning*, Seoul, South Korea. PMLR 306, 2026. Copyright 2026 by the author(s).

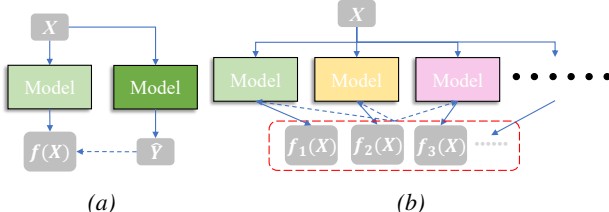

*Figure 1.* (a) Prior LiDAR SemiSL methods (Kong et al., 2023c; Chen et al., 2021b; Li et al., 2023; Liu et al., 2024) adopt a decoupled two-step design, where pseudo-labels are generated from a single source of LiDAR representation for supervision. (b) *CoLLiS* trains multiple LiDAR representations collaboratively in a single step, treating all models as coequal students and enabling adaptive knowledge transfer from all participating representations.

tion. These approaches predominantly leverage range-view images (Milioto et al., 2019; Ando et al., 2023), voxel grids (Zhu et al., 2021; Choy et al., 2019) and polar images (Zhang et al., 2020). To combine complementary advantages of various LiDAR geometry in perception, prior works have progressively focused on fusing different LiDAR representations (Xu et al., 2021; Hou et al., 2022).

Despite these advancements, existing methods primarily rely on fully supervised training, requiring vast amounts of fine-grained annotated data to achieve baseline accuracy. Annotating large-scale LiDAR datasets, however, is prohibitively time-consuming and labor-intensive (Abdelsamad et al., 2025; 2026). Such limitations have driven the exploration into semi-supervised learning (SemiSL) for LiDAR semantic segmentation, where models are trained on limited labeled data alongside abundant unlabeled samples. Early LiDAR SemiSL approaches adopt consistency regularization under input perturbations within two-step frame-

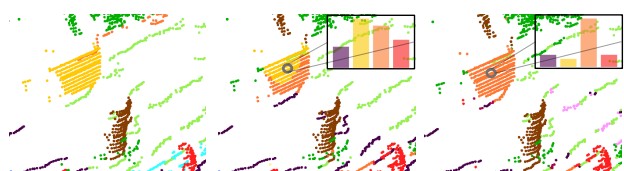

*Figure 2.* Confirmation bias: distillation from a single source can over-fit to its own errors. From left to right: ground-truth labels, predictions at $t_1$, and predictions at $t_2$ ($t_1 < t_2$).

works (Kong et al., 2023c; Li et al., 2023; Unal et al., 2022), where predictions from weakly augmented inputs serve as pseudo-labels to supervise strongly augmented ones. More recent works extend this paradigm to dual-representation settings (Liu et al., 2024), in which pseudo-labels are generated from a different LiDAR representation to exploit complementary geometric cues. Leveraging such complementary representations can improve generalization by capturing invariant information across views (Rath & Condurache, 2020; Coors et al., 2018). For example, range-view representations are dense but suffer from projection distortions (Yang et al., 2024), whereas voxel-based representations preserve regular spatial structure but may struggle in sparse regions (Xu et al., 2021).

However, existing SemiSL methods remain vulnerable to **confirmation bias** (Arazo et al., 2020), where erroneous pseudo-labels are reinforced during training, especially under high label scarcity. This issue is exacerbated in LiDAR segmentation due to the long-tail distribution of objects and the architectural biases of individual networks. Although dual-representation approaches can alleviate this problem to some extent by introducing complementary views, they still rely on a single pseudo-labeling source during training. Consequently, models may overfit to noisy pseudo-labels, leading to limited generalization.

To address this challenge, we propose *CoLLiS*, a novel collaborative framework for semi-supervised LiDAR semantic segmentation. *CoLLiS* treats multiple LiDAR representations as coequal student models that learn collaboratively in a single forward step. Through adaptive distillation with multiple representations, our framework regularizes the over-reliance on a single pseudo-labeling source, thereby improving the generalization of each participating model. Moreover, the streamlined single-step design effectively balances the efficiency and scalability when extending to multiple representations. To further enhance collaborative learning, we introduce a dynamic augmentation mechanism that adaptively controls augmentation intensity based on consensus across multiple representations. We summarize our main contributions as follows:

1. We introduce a **consensus-driven augmentation mechanism** that dynamically adjusts augmentation strength based on peer agreement, thereby enhancing generalization in SemiSL. (Sec. 3.2.1)
2. We propose an **adaptive pseudo-labeling and distillation** strategy that balances online knowledge transfer across multiple models by accounting for inter-student disparities (Sec. 3.2.2).
3. We conduct extensive experiments on three widely used semi-supervised LiDAR benchmarks under diverse training settings to validate the effectiveness of the proposed framework.

## 2. Related works

**LiDAR segmentation and SemiSL** LiDAR semantic segmentation methods differ by input representation: some process raw point clouds (Wu et al., 2024; Puy et al., 2023), while others impose structure through range/polar projections (Zhao et al., 2021; Zhang et al., 2020) or voxel grids (Zhu et al., 2021; Choy et al., 2019). Multi-view fusion leverages complementary cues (Hou et al., 2022; Xu et al., 2021), but all require costly large-scale annotations. Semi-supervised learning (SemiSL) reduces this burden by exploiting unlabeled data. Recent works adopt consistency regularization with LiDAR-specific perturbations (Kong et al., 2023c; Xu et al., 2023) or contrastive learning (Li et al., 2023; Li & Dong, 2024; Liu et al., 2024). Yet most methods rely on two-step pipelines, limited to one or two representations, and still suffer from confirmation bias due to distillation from a single model. We address this challenge with a one-step collaborative framework that integrates multiple LiDAR representations for scalable and robust semi-supervised learning. Notably, our framework explicitly focuses on model collaboration under label scarcity. We empirically show that it is complementary to existing LiDAR SemiSL approaches that primarily focus on consistency regularization (see Sec. 4.4).

**Collaborative learning (CoL)** eliminates the need for explicit pseudo-labeling by enabling peer networks to learn from each other. DML (Zhang et al., 2018) pioneered reciprocal supervision between two networks, while KDCL (Guo et al., 2020) and ONE (Zhu et al., 2018) introduced ensemble-based pseudo-teachers. More recent studies (Liu et al., 2022; Zhu et al., 2023) extend this idea to heterogeneous architectures, such as CNNs and Vision Transformers (Dosovitskiy et al., 2021), demonstrating enhanced generalization through collaboration among heterogeneous models. Despite these successes, CoL remains underexplored in semi-supervised settings and largely absent in the LiDAR domain, where sparse geometry and noisy pseudo-labels pose unique challenges.

## 3. *CoLLiS*

### 3.1. Preliminaries

We consider a LiDAR point cloud with $N$ points $\mathbf{P} = p_i \mid p_i = (x, y, z, I)_i$, where $(x, y, z)$ are coordinates and $I$ is intensity. Training data consists of a labeled set $D_l = (\mathbf{P}_j^l, Y_j^l)$ with one-hot labels $Y_j^l \in \mathbb{R}^{N \times K}$ for $K$ classes, and an unlabeled set $D_u = \mathbf{P}_j^u$, with $|D_l| \leq |D_u|$. To impose geometric priors, $\mathbf{P}$ is typically transformed into structured forms such as range images $\mathbf{R} \in \mathbb{R}^{U \times V \times C_r}$ via spherical projection (Milioto et al., 2019) or voxel grids $\mathbf{V} \in \mathbb{R}^{H \times W \times L \times C_v}$ via discretization (Zhu et al., 2021). Since these representations differ in structure, we use the

point cloud as an intermediary to define cross-representation mappings, i.e., range-to-voxel and voxel-to-range transformations can be expressed as $T_{r \to v} = T_{p \to r}^{-1} \circ T_{p \to v}$ and $T_{v \to r} = T_{p \to v}^{-1} \circ T_{p \to r}$. These mappings are essential for enabling effective distillation across different representations.

## 3.2. Pipeline

### 3.2.1. CONSENSUS-DRIVEN AUGMENTATION

In semi-supervised learning, limited annotations restrict generalization and undermine pseudo-label quality. Collaborative paradigms require both diverse training data to improve generalization and reliable pseudo-labels to guide distillation. When augmentation intensity is fixed, they become prone to under- or over-fitting.

A naive solution is to adjust the probability using curriculum learning (CL) (Bengio et al., 2009), where augmentation strength gradually increases with training progression. However, CL requires sensitive hyperparameter tuning and enforces a rigid schedule, which is suboptimal. To address this, we propose a consensus-driven augmentation (CDA) mechanism that automatically adjusts the mixing probability ($q_m$) based on inter-student consistency. Importantly, our focus is on controlling augmentation intensity dynamically rather than designing new mixing methods. We detail this mechanism below.

We first compute a fraction $a_n$ of predictions that are consistent across students over a step size $N$:

$$a_n = \frac{\sum_{i,j \in \{1,2,3\}}^{i \neq j} \mathbb{I}(\hat{Y}_{n,s_i} = \hat{Y}_{n,s_j})}{\sum \mathbb{I}(\hat{Y}_n)}, \qquad (1)$$

where $\mathbb{I}(\cdot)$ is the indicator function. The ratio acts as a measure of data complexity: when students agree, the sample is likely easy to learn and can benefit from stronger augmentations to increase diversity. Conversely, when students disagree, the sample is inherently harder, so weaker augmentations are applied to avoid introducing additional noise. Next we transform these ratios based on whether the point clouds have been mixed:

$$g(a) = \begin{cases} a - 1, & \text{if mixed.} \\ a, & \text{otherwise.} \end{cases} \qquad (2)$$

Finally, the mixing probability is updated iteratively based on the past probability:

$$q_{m,t} = q_{m,t-1} * (1 + \sum_{n=1}^{N} g(a_n)), \qquad (3)$$

In practice, we observe that CDA consistently outperforms CL in both accuracy and robustness (see Sec. 4.4 for supporting experiments). To further enhance scene diversity, we integrate multiple mixing strategies from prior works: Laser-Mix (Kong et al., 2023c), PolarMix (Xiao et al., 2022), and Sub-cloud Shuffling (Yang & Condurache, 2026), with one strategy randomly selected at each iteration during training (see Appendix A.5 for further details).

### 3.2.2. TRAINING PROCEDURE

**On-the-fly pseudo-labeling** In *CoLLiS*, each student treats all other participants as potential sources of pseudo-labels to avoid over-reliance on a single source. However, naive distillation from multiple representations can degrade performance due to contradictory pseudo-labels. To address this issue, we selectively filter supervision and perform adaptive distillation across representations. The selection is guided by two factors: *Absolute Reliability (AR)* and *Relative Reliability (RR)*. AR quantifies a model's intrinsic confidence as training progresses, while RR calibrates reliability by comparing peers. Together, these metrics enhance training robustness by down-weighting unreliable information and balancing knowledge transfer among students of varying strengths.

Motivated by empirical findings that pseudo-label reliability improves as training proceeds (Wang et al., 2022), we model AR (denoted as $\beta$) as a linear function of the training epoch and use it to compute the weight of the unlabeled loss ($\lambda_u$):

$$\beta(e) = \frac{e}{E_{\max}}, \quad \lambda_u = \lambda_0(1 - \beta) + \beta, \qquad (4)$$

where $e$ is the current epoch, $E_{\max}$ is the total number of training epochs, and $\lambda_0$ is the initial loss weight.

To establish the relative reliability ($RR$, denoted as $\gamma$), we leverage prediction confidence as a proxy for uncertainty estimation. From a Bayesian perspective, an ideal measurement of reliability for a student model $s$ with parameters $\omega$ at input $x$ is the posterior probability of correctness $p(y_s(x) = y | x, \omega)$, where $y_s(x)$ denotes the prediction and $y$ the ground-truth label. In practice, this quantity is hardly accessible. As neural networks are known to be imperfectly calibrated (Guo et al., 2017), this means that their predicted uncertainty $\max_y p(y | x, \omega)$ does not accurately reflect the true probability of correctness. Consequently, the models may exhibit overconfidence or underconfidence, which can be further exacerbated in semi-supervised setting. However, provided that models are trained with a non-trivial amount of labeled data, calibration errors predominantly affect the magnitude of confidence scores rather than their relative ordering (Ovadia et al., 2019). As a result, while the absolute value of confidence may be inaccurate, relative confidence comparisons across models remain informative. In particular, when one model consistently produces higher-confidence predictions than another on the same inputs, its predictions tend to be more trustworthy. Based on this observation, we define relative reliability by comparing con-

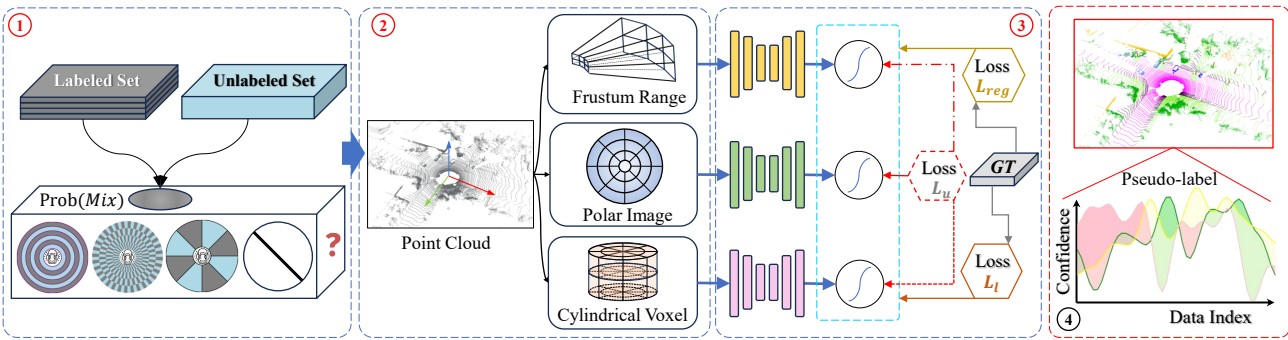

*Figure 3.* Overview of *CoLLiS*. ① The labeled dataset $D_l$ is repeated to match the size of the unlabeled dataset $D_u$. A batch from each is sampled and optionally mixed using a random mixing strategy with non-fixed probability. ② The sampled point clouds are transformed into multiple LiDAR representations. ③ Each model is trained with a composite loss consisting of a labeled loss ($L_l$), a regularization term ($L_{reg}$), and an unlabeled loss ($L_u$). Pseudo-labels are generated online using confidence-based modeling. ④ Confidence curves produced by all student models over the same inputs. Relative reliability $\gamma$ is estimated by counting the frequency of confidence dominance across inputs, corresponding to the colored regions where one model exhibits higher confidence than the others.

fidence rankings between students. Specifically, we quantify the reliability of student $s_1$ relative to student $s_2$ by estimating the frequency with which $s_1$ exhibits higher confidence than $s_2$, as expressed below:

$$\gamma_{s_1 \to s_2} = \frac{N_{s_1 \to s_2}}{N_{s_2 \to s_1}}, N_{s_i \to s_j} = \sum \mathbb{I}(c(P_{s_i}) > c(P_{s_j})) \tag{5}$$

$c(\cdot)$ is the confidence of prediction and $N_{s_i \to s_j}$ is the count of points for which confidence of student $i$ exceeds that of student $j$. $\mathbb{I}(\cdot)$ denotes the indicator function. This pairwise counting formulation serves as a robust, rank-based estimate of relative uncertainty. Compared with mean-confidence measures and entropy-based weighting, it is less sensitive to both overconfidence spikes in maximum confidence and averaging effects such as oversmoothing.

Next, the threshold value $\delta$ and the pseudo-labels are obtained by Eq. 6 and Eq. 7:

$$\delta(t)_{s_1 \to s_2} = min(\delta_0, \delta_0 * ((1 - \beta) * \frac{1}{\gamma_{s_1 \to s_2}})) \tag{6}$$

$$\hat{Y}_{s_1 \to s_2} = \{argmax(P_{s_1}) | c(P_{s_1}) > \delta(t)_{s_1 \to s_2}\}, \tag{7}$$

where $\delta_0$ is the predefined maximum and $\hat{Y}_{s_i \to s_j}$ are pseudo-labels from student $i$ to $j$. Specifically, high absolute and relative reliability will result in the lower threshold value, leading to more tolerant pseudo-labeling and increase in distillation power. We present the results regarding the impact of our reliability modeling on pseudo-label quality in Sec. 4.3. All filtered pseudo-labels are subsequently mapped back to model-specific representations to preserve geometric consistency.

**Labeled loss** The labeled loss is computed using the ground-truth annotations:

$$L_l = \mathcal{L}(P_s, Y), \tag{8}$$

where $P_s$ denotes the prediction of student $s$ and $Y$ is the ground-truth label. Following prior work (Liu et al., 2024; Kong et al., 2023c), the loss function $\mathcal{L}$ is defined as the sum of cross-entropy loss and Lovász loss (Berman et al., 2018).

**Unlabeled loss** As label conflicts may still arise because each student filters pseudo-labels independently using its own confidence threshold, we further introduce adaptive weighting to softly resolve such conflicts during distillation. The weights are derived from the normalized confidence counts, which estimate how frequently one student exhibits lower predictive uncertainty than another. This yields a data-driven approximation of source reliability during distillation. Specifically, for knowledge transfer from students $s_1$ and $s_2$ to a target student $s_3$, the weights are defined as:

$$\omega_{s_1 \to s_3} = \frac{N_{s_1 \to s_2}}{N_{s_2 \to s_1} + N_{s_1 \to s_2}}, \quad \omega_{s_2 \to s_3} = 1 - \omega_{s_1 \to s_3}. \tag{9}$$

Finally, the unlabeled loss for each student (student 1 as an example) is computed as:

$$L_{u, s_1} = \lambda_u \cdot \sum_{s_i \neq s_1} \omega_{s_i \to s_1} \cdot \mathcal{L}(P_{s_1}, \hat{Y}_{s_i \to s_1}) \tag{10}$$

where $\mathcal{L}$ is the same loss function as in $L_l$.

**Regularization loss** Because *CoLLiS* relies on prediction confidence for pseudo-labeling and distillation, this assumption may become vulnerable under severe distribution shifts or extreme label scarcity. We therefore further add a regularization loss (Zou et al., 2019) to safeguard against such edge cases:

$$L_{reg} = -\lambda_{reg} \sum_{k=1}^{K} \frac{1}{K} \log P(k), \tag{11}$$

where $K$ is the number of classes, $P(k)$ the softmax probability, and $\lambda_{reg} = 0.1$. This KL divergence to a uniform distribution smooths predictions and prevents overconfidence.

# 4. Experiments & Discussion

## 4.1. Experimental setup

We evaluate *CoLLiS* on three LiDAR segmentation benchmarks: nuScenes (Fong et al., 2022), SemanticKITTI (Behley et al., 2019) and ScribbleKITTI (Unal et al., 2022). For all three datasets, we follow the settings of previous works (Kong et al., 2023c; Liu et al., 2024): uniformly sampling 1%, 10%, 20% and 50% labeled data for training and the rest of dataset as unlabeled set. In the default setting of *CoLLiS*, we employ FRNet (Xu et al., 2023) to process frustum-range-view representation, PolarNet (Zhang et al., 2020) for bird's-eye-view images, and Cylinder3D (Zhu et al., 2021) for voxel representation. More implementation details are provided in Appendix A.2 and A.1 to ensure the full reproducibility.

## 4.2. Comparative study

**Uniform splits** We evaluate *CoLLiS* against existing LiDAR SemiSL approaches across diverse input representations, datasets, and label proportions (Tab. 1). Compared with single-representation methods such as LaserMix (Kong et al., 2023c) and FrustumMix (Xu et al., 2023), *CoLLiS* consistently achieves higher performance across most settings. The advantages are most evident in annotation-scarce scenarios, where confirmation bias is particularly severe: *CoLLiS* delivers strong gains at 1% and 10% label ratios on nuScenes (Fong et al., 2022) and SemanticKITTI (Behley et al., 2019), while also achieving consistent improvements on the sparsely annotated ScribbleKITTI (Unal et al., 2022). These results demonstrate that collaborative learning across multiple representations effectively mitigates confirmation bias and achieves substantial advances in LiDAR SemiSL.

**Significant splits** Additionally, we compare our method with prior works that adopt a significant data split, where frame overlap is minimized (Tab. 2). All compared methods use Cylinder3D as the backbone. To ensure a fair comparison, we reproduce our results using an increased voxel resolution to match their hyperparameter settings. Under this configuration, *CoLLiS* consistently outperforms state-of-the-art approaches.

**Qualitative results** Additionally, we provide the qualitative results in Fig. 4. Compared with IT2, *CoLLiS* shows fewer scattered errors across all three scenes. The improvements are especially visible in sparse regions and around object boundaries, where IT2 produces larger clusters of incorrect predictions. In addition, *CoLLiS* produces more

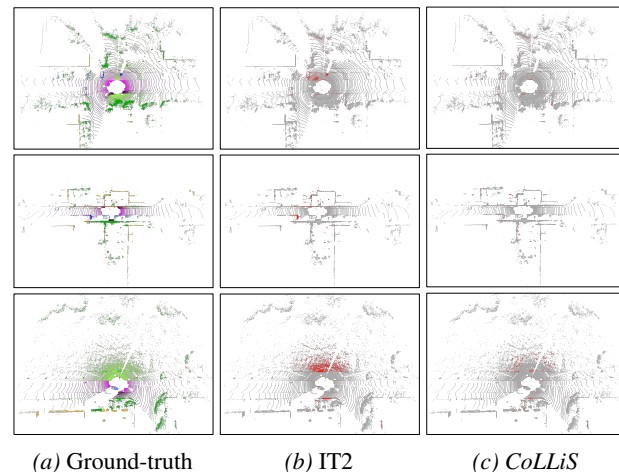

*(a)* Ground-truth          *(b)* IT2          *(c)* *CoLLiS*

*Figure 4.* We qualitatively evaluate Cylinder3D (Zhu et al., 2021) on SemanticKITTI. Predictions are obtained from models trained under 10% label protocol. Ground-truth labels are color-coded based on class categories. Incorrect predictions are shown in red, while correct predictions are shown in gray. More qualitative results are provided in Appendix C.2.

spatially consistent predictions with fewer isolated misclassified points, indicating improved robustness under limited supervision.

## 4.3. Effectiveness of Collaboration

Beyond performance gain compared to existing approaches, we further analyze the effectiveness of our method from various perspectives, with a particular focus on verifying whether the proposed collaborative paradigm is necessary for training with multiple LiDAR representations in semi-supervised context.

**Quality of pseudo-labels** We evaluate pseudo-label quality using their retention rate and accuracy during training. The retention rate reflects whether supervision collapses toward a single dominant model, while accuracy measures the correctness of retained pseudo-labels. As shown in Fig. 5, all representations preserve a non-trivial fraction of pseudo-labels after thresholding, and the overall retention rate increases over time, indicating improving pseudo-label reliability. Although stronger representations retain more pseudo-labels, weaker ones are not eliminated. Since thresholding is applied independently to each student, less dominant models can still contribute reliable pseudo-labels in regions where they are confident. This prevents collaboration from collapsing to a single modality and keeps all representations actively involved.

**Mitigating confirmation bias** For comparison, we adopt a standard collaborative learning baseline with naive mutual distillation. As shown in Fig. 6a, the baseline exhibits a clear performance collapse midway through training, followed

*Table 1.* Comparison with the state-of-the-art LiDAR SemiSL methods. All representations are evaluated with standalone architectures during inference without test time augmentation or model ensemble. The **best** and second best result for each setting of label proportion is highlighted in **bold** and underline. $\star$: reproduced results using the released codebase. IT2$_R$ and IT2$_V$ denote models trained together with range-view and voxel representations, respectively. Class-wise performance is reported in Appendix C.1

| Repr. | Method | nuScenes | | | | SemanticKITTI | | | | ScribbleKITTI | | | |
|---|---|---|---|---|---|---|---|---|---|---|---|---|---|
| | | 1% | 10% | 20% | 50% | 1% | 10% | 20% | 50% | 1% | 10% | 20% | 50% |
| *F-Range* | sup. only | 51.9 | 68.1 | 70.9 | 74.6 | 44.9 | 60.4 | 61.8 | 63.1 | 42.4 | 53.5 | 55.1 | 57.0 |
| | PolarMix (Xiao et al., 2022) | 55.6 | 69.6 | 71.0 | 73.8 | 50.1 | 60.9 | 62.0 | 63.8 | 43.2 | 55.0 | 56.1 | 57.3 |
| | LaserMix (Kong et al., 2023c) | 58.7 | 71.5 | 72.3 | 75.0 | 52.9 | 62.9 | 63.2 | 65.0 | 45.8 | 56.8 | 57.7 | 59.0 |
| | FrustumMix (Xu et al., 2023) | 61.2 | 72.2 | 74.6 | 75.4 | 55.8 | **64.8** | **65.2** | 65.4 | 46.6 | 57.0 | 59.5 | **61.2** |
| | *CoLLiS* (Ours) | **63.2** | **74.2** | **74.8** | **75.8** | **56.0** | 64.3 | 64.9 | **66.2** | **47.6** | **59.9** | **60.1** | 60.7 |
| *Polar* | sup. only | *46.5 | 58.5 | 63.9 | 68.4 | *41.6 | *50.2 | *51.8 | *53.0 | *36.2 | *46.5 | *48.1 | *49.6 |
| | LaserMix (Kong et al., 2023c) | *53.6 | *64.5 | *66.5 | *69.3 | - | - | - | - | - | - | - | - |
| | IT2$_R$ (Liu et al., 2024) | *52.9 | 64.8 | 67.9 | 70.6 | - | - | - | - | - | - | - | - |
| | IT2$_V$ (Liu et al., 2024) | *54.4 | 66.3 | 69.1 | 71.6 | - | - | - | - | - | - | - | - |
| | *CoLLiS* (Ours) | **57.9** | **68.4** | 68.6 | 70.8 | **46.8** | **53.3** | **54.0** | **55.5** | **42.0** | **51.1** | **51.4** | **51.7** |
| *Voxel* | sup. only | 50.9 | 65.9 | 66.6 | 71.2 | 45.4 | 56.1 | 57.8 | 58.7 | 39.2 | 48.0 | 52.1 | 53.8 |
| | CBST (Zou et al., 2018) | 53.0 | 66.5 | 69.6 | 71.6 | 48.8 | 58.3 | 59.4 | 59.7 | 41.5 | 50.6 | 53.3 | 54.5 |
| | CPS (Chen et al., 2021b) | 52.9 | 66.3 | 70.0 | 72.5 | 46.7 | 58.7 | 59.6 | 60.5 | 41.4 | 51.8 | 53.9 | 54.8 |
| | LaserMix (Kong et al., 2023c) | 55.3 | 69.9 | 71.8 | 73.2 | 50.6 | 60.0 | 61.9 | 62.3 | 44.2 | 53.7 | 55.1 | 56.8 |
| | IT2 (Liu et al., 2024) | 57.5 | 72.0 | **73.6** | 74.1 | 52.0 | 61.4 | 62.1 | 62.5 | 47.9 | 56.7 | 57.5 | 58.3 |
| | *CoLLiS* (Ours) | **61.1** | **72.9** | 73.4 | **74.5** | **53.2** | **63.1** | **63.6** | **64.0** | 47.6 | **58.6** | **58.8** | **59.0** |

*Table 2.* Results on significant data split.

| Method | nuScenes | | SemanticKITTI | |
|---|---|---|---|---|
| | 1% | 10% | 1% | 10% |
| GPC (Jiang et al., 2021) | - | - | 54.1 | 62.0 |
| Lim3D (Li et al., 2023) | - | - | 58.4 | 62.2 |
| DDSemi (Li & Dong, 2024) | 58.1 | 70.2 | 59.3 | 65.1 |
| AIScene (Liu et al., 2025) | 60.2 | 72.3 | 61.2 | 66.3 |
| *CoLLiS* (Voxel) | **63.1** | **74.5** | **61.5** | **67.0** |

where $N^\star$ is the number of incorrect pseudo-labels, $P^\star$ is their predicted probability distribution and $\mathbf{U}$ is a uniform prior. This metric reflects how confident a model is when it makes incorrect predictions. We track this value throughout training and report the results in Fig. 6b. As shown, *CoLLiS* consistently reduces the certainty of incorrect predictions, whereas this certainty steadily increases when models are trained under the standard collaborative learning setup.

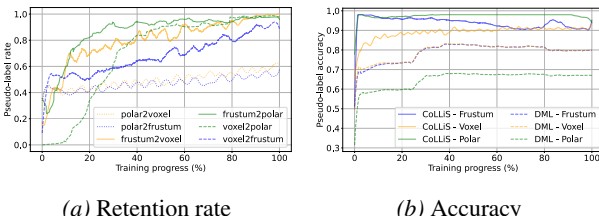

*(a)* Retention rate      *(b)* Accuracy

*Figure 5.* Quality of pseudo-labels. Values are averaged over every 500 iterations, and training is conducted on nuScenes (Fong et al., 2022) with 1% labeled data.

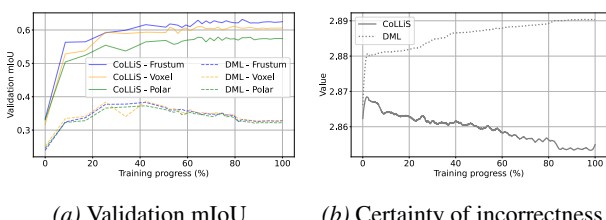

*(a)* Validation mIoU      *(b)* Certainty of incorrectness

*Figure 6.* We monitor two metrics over training and compare with standard collaborative learning paradigm (DML (Zhang et al., 2018)). Values are averaged over every 500 iterations, and training is conducted on nuScenes (Fong et al., 2022) with 1% labeled data.

by continuous degradation across all representations. This behavior indicates that conventional collaborative learning struggles to recover once representation drift occurs. In contrast, *CoLLiS* maintains stable and consistent performance improvements across all representations throughout training. Furthermore, following prior work (Arazo et al., 2020), we directly measure confirmation bias to quantitatively assess whether our method mitigates this issue. Specifically, we measure the average certainty of incorrect predictions as

$$-\frac{1}{N^\star} \sum_{n=1}^{N^\star} \mathbf{U}^T \log\left(P^\star\right), \qquad (12)$$

**Training efficiency** Since *CoLLiS* employs multiple models for collaborative learning, evaluating its training efficiency is essential. Tab. 3 summarizes the training time and memory consumption. Compared with IT2 (Liu et al., 2024), *CoLLiS* requires substantially less memory and achieves approximately 50% faster training. This efficiency advantage stems directly from our streamlined single-step design, which avoids additional time-consuming components such as contrastive loss computation required by IT2. Even relative to LaserMix (Kong et al., 2023c), a single-representation approach, *CoLLiS* exhibits comparable memory usage.

*Table 3.* **Training efficiency**. The backnone of LaserMix is Cylinder3D (Zhu et al., 2021). The training time is measured on a single H200 GPU. Measurement of *CoLLiS* is the sum of all collaborative models.

| Method | Training time | Memory |
|---|---|---|
| LaserMix (Kong et al., 2023c) | 15.4 h | 10.2G |
| IT2 (Liu et al., 2024) | 50.8 h | 13.11G |
| *CoLLiS* | 25.6 h | 11.06G |

## 4.4. Ablation study

**Scalability with more representations** To assess how well the framework scales with additional collaborators, we introduce a fourth representation using Point Transformer V3 (PTv3) (Wu et al., 2024) and train *CoLLiS* with varying numbers of representations. We fix the label ratio at 1%, where PTv3 also underperforms, to more rigorously evaluate the effectiveness of *CoLLiS* under limited supervision. As shown in Tab. 4, incorporating additional representations yields consistent performance gains across all models. We observe that improvements for stronger representations gradually saturate as more collaborators are introduced. This behavior is expected, as strong models already capture most discriminative patterns under limited supervision, and additional collaborators mainly provide redundant or confirmatory pseudo-labels. Importantly, their performance remains stable, while weaker representations benefit substantially from collaboration. These results highlight the scalability and robustness of *CoLLiS* when extending to multiple LiDAR representations.

**Scalability with existing frameworks** We observe that *CoLLiS* is highly compatible with existing semi-supervised LiDAR frameworks due to its simple and modular design. To demonstrate this, we integrate *CoLLiS* with IT2 (Liu et al., 2024). In its original formulation, IT2 supervises each representation using pseudo-labels generated exclusively from the other representation. We extend this setup by incorporating both pseudo-labels while applying our adaptive pseudo-labeling and distillation strategy. The results are reported in Tab. 5, where we also include the performance of

*Table 4.* We evaluate the scalability of *CoLLiS* by training it with varying numbers of representations, and we report the performance of each model independently.

| #Repr. | FRNet (F-Range) | PolarNet (Polar) | Cylinder3D (Voxel) | PTv3 (Raw points) |
|---|---|---|---|---|
| nuScenes 1% | | | | |
| sup. only | 51.9 | 46.5 | 50.9 | 48.0 |
| 2 | 62.5 | - | 59.2 | - |
| 3 | 63.2 | 57.9 | 61.1 | - |
| 3 | **63.5** | - | 61.3 | 60.7 |
| 4 | 63.4 | **58.6** | **61.6** | **61.2** |
| SemanticKITTI 1% | | | | |
| sup. only | 44.9 | 41.6 | 45.4 | 41.6 |
| 2 | 55.1 | - | 51.8 | - |
| 3 | 56.0 | 46.8 | 53.2 | - |
| 3 | 56.8 | - | 54.3 | 54.8 |
| 4 | **56.8** | **48.9** | **54.7** | **55.2** |

individual baselines for reference. Combining *CoLLiS* with IT2 leads to clear and consistent performance gains, highlighting the complementary strengths of the two approaches. In particular, IT2's contrastive objective and weak-to-strong view consistency enhance representation alignment, while our collaborative learning paradigm mitigates confirmation bias by introducing additional pseudo-labeling sources for each representation.

*Table 5.* Combination of *CoLLiS* and IT2 (Liu et al., 2024)

| Model | FIDNet | | Cylinder3D | |
|---|---|---|---|---|
| | 1% | 10% | 1% | 10% |
| IT2 | 56.5 | 71.2 | 57.5 | 72.1 |
| *CoLLiS* | 57.2 | 71.4 | 57.4 | 71.2 |
| IT2 + *CoLLiS* | **58.4** | **72.5** | **60.8** | **73.1** |

**Component designs** Tab. 6 presents the ablation of *CoLLiS* on FRNet (Xu et al., 2023) and Cylinder3D (Zhu et al., 2021) under 1% and 20% label protocols. With 1% labels, both models gain substantially (FRNet: +4.7%, Cylinder3D: +3.3%), but benefits diminish at 20%. Incorporating two adaptive reliability factors improves robustness across two label ratios significantly, while confidence regularization (Zou et al., 2019) and mixing-based augmentation add further gains (up to +3.0%). Scaling collaboration to three representations with polar images (Zhang et al., 2020) yields additional improvements.

*Table 6.* Full ablation on the nuScenes (Fong et al., 2022) dataset. Starting from the results of supervised training, we regard the mutual distillation with two representations as the baseline. AR and RR denote absolute and relative reliability for pseudo-labeling. CDA represents consensus-driven augmentation.

| Co. | AR | RR | $L_{reg}$ | CDA | +Polar | FRNet | | Cylinder3D | |
|---|---|---|---|---|---|---|---|---|---|
| | | | | | | 1% | 20% | 1% | 20% |
| sup. only | | | | | | 51.9 | 70.9 | 50.9 | 66.6 |
| ✓ | | | | | | 56.6 | 71.0 | 54.2 | 66.2 |
| ✓ | ✓ | | | | | 57.5 | 71.6 | 55.5 | 67.4 |
| ✓ | ✓ | ✓ | | | | 58.8 | 72.3 | 56.5 | 69.0 |
| ✓ | ✓ | ✓ | ✓ | | | 59.5 | 72.8 | 57.3 | 70.4 |
| ✓ | ✓ | ✓ | ✓ | ✓ | | 62.5 | 74.4 | 59.2 | 73.1 |
| ✓ | ✓ | ✓ | ✓ | ✓ | ✓ | 63.2 | 74.8 | 61.1 | 73.4 |

**Dynamic augmentation** As shown in Fig. 7 (left), both CL and fixed $q_m$ are highly sensitive to hyper-parameter initialization, requiring careful tuning to avoid performance degradation. In contrast, CDA maintains stable performance across diverse initialization settings, demonstrating parameter-agnostic adaptation. Its robustness arises from dynamically adjusting mixing intensity based on inter-student consensus, which enables reliable and adaptive training.

We further examine the effect of step size in CDA (Fig. 7, right). A step size of 50 yields the best performance, indicating that frequent adjustments of augmentation intensity help prevent overfitting to a fixed perturbation level and

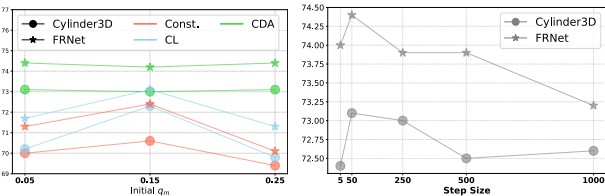

*Figure 7.* Ablation study on initialization of mixing probability $q_m$ (left) and step size of CDA (right) with nuScenes (20% labels).

thus improve generalization. However, excessively small step sizes risk biased updates due to insufficient samples. Setting the step size to 50 balances training stability with the flexibility of dynamic augmentation.

**Adaptive weights** As shown in Tab. 7, adaptive weights yield clear improvements over fixed weights, demonstrating their effectiveness in reducing noise caused by contradictory pseudo-labels from different representations.

*Table 7.* Ablation study on the adaptive distillation weights ($\omega_{s_i \rightarrow s_j}$). We evaluated Cylinder3D (Zhu et al., 2021) on nuScenes (Fong et al., 2022) dataset with 1% and 10% labels.

| $\omega_{s_i \rightarrow s_j}$ | FRNet | | Cylinder3D | |
|---|---|---|---|---|
| | 1% | 20% | 1% | 20% |
| Fixed = 1 | 61.7 | 72.5 | 59.0 | 71.5 |
| Adaptive | 63.2 | 74.8 | 61.1 | 73.4 |

**Weight of unlabeled loss** We further examine the effect of initial unlabeled loss weights in Tab. 8. An improper choice consistently degrades segmentation performance across all student models. Setting the weight too high causes overfitting to incorrect pseudo-labels early in training, while setting it too low prevents sufficient knowledge transfer, leading to overfitting on the limited labeled data. This underscores the importance of careful tuning to balance knowledge transfer and stability in *CoLLiS*. We also evaluate adaptive distillation weights in Tab. 7. The results show clear performance gains, confirming that softly resolving label conflicts is crucial for robust collaborative training.

*Table 8.* Ablation study on the initial weight of unlabeled loss ($\lambda_0$). Results are reported on nuScenes (Fong et al., 2022) dataset with 10% labels.

| $\lambda_0$ | FRNet | PolarNet | Cylinder3D |
|---|---|---|---|
| 0.2 | 73.5 | 66.9 | 72.1 |
| 0.5 | **74.2** | **68.4** | **72.9** |
| 0.8 | 73.1 | 67.4 | 72.3 |

**Dynamic augmentation** Tab. 9 compares adaptation strategies for mixing probability. Curriculum Learning (CL) outperforms fixed $q_m$ under moderate label scarcity (20%) but loses effectiveness under extreme scarcity (1%). In contrast, our consensus-driven augmentation (CDA) consis-

tently achieves the best results across both label regimes and networks. For mixing strategies, combining LaserMix (*LM*), PolarMix (*PM*), and Sub-cloud Shuffling (*SS*) yields the strongest performance, as each exploits complementary geometric cues: vertical beams, radial sectors, and global scene integrity.

*Table 9.* Ablation study on dynamic augmentation with nuScenes (Fong et al., 2022) dataset. Const. denotes that a constant value is assigned to $q_m$ (0.25 for 1% and 0.15 for 20%). For curriculum learning (CL), $(q_{m,min}, q_{m,max})$ is set to (0.2, 0.3) for 1% and (0.15, 0.25) for 20%, respectively. The step size ($N$) for CDA is fixed at 50 for both settings. {*LM, PM, SS*} are three different mixing strategies.

| Const. | CL | CDA | *LM* | *PM* | *SS* | FRNet | | Cylinder3D | |
|---|---|---|---|---|---|---|---|---|---|
| | | | | | | 1% | 20% | 1% | 20% |
| ✓ | | | ✓ | ✓ | ✓ | 61.1 | 72.4 | 58.0 | 70.6 |
| | ✓ | | ✓ | ✓ | ✓ | 60.9 | 73.1 | 58.2 | 72.3 |
| | | ✓ | ✓ | ✓ | ✓ | **62.5** | **74.4** | **59.2** | **73.1** |
| | ✓ | | ✓ | | | 60.4 | 73.2 | 58.2 | 72.1 |
| | ✓ | | ✓ | ✓ | | 61.8 | 74.4 | 59.0 | 72.9 |

## 4.5. Other settings

**Out-of-Distribution scenario** We additionally evaluate each individual model after collaborative training on Robo3D (Kong et al., 2023b), which contains data simulated under various corruption scenarios. Although confidence-based strategies may be less reliable under severe distribution shifts, *CoLLiS* achieves performance comparable to its baseline counterparts, as shown in Tab. 10. This result indicates that leveraging multiple representations during training improves robustness to out-of-distribution perturbations.

*Table 10.* Results on nuScenes-C (Robo3D (Kong et al., 2023b)). In addition to mCE, mRR, and mIoU, we also report the IoU for each corruption type. [†] indicates models trained with *CoLLiS*.

| Model | mCE ↓ | mRR ↑ | mIoU↑ | Fog | Weg | Snow | Motion | Beam | Cross | Echo | Sensor |
|---|---|---|---|---|---|---|---|---|---|---|---|
| FRNet | 98.6 | 77.5 | 77.7 | 69.1 | 76.6 | 69.5 | 54.5 | 68.3 | 41.4 | 58.7 | 43.1 |
| FRNet[†] | 99.8 | 77.0 | 77.1 | 68.2 | 76.9 | 69.0 | 51.8 | 66.9 | 43.6 | 59.2 | 42.5 |
| Cylinder3D | 111.8 | 72.9 | 76.2 | 59.9 | 72.7 | 58.1 | 42.1 | 64.5 | 44.4 | 60.5 | 42.2 |
| Cylinder3D[†] | 110.1 | 74.1 | 76.8 | 61.2 | 70.5 | 63.2 | 43.7 | 65.2 | 42.3 | 61.0 | 41.0 |
| PolarNet | 115.1 | 76.3 | 71.4 | 58.2 | 69.9 | 64.8 | 44.6 | 61.9 | 40.8 | 53.6 | 42.0 |
| PolarNet[†] | 111.5 | 77.9 | 74.3 | 60.4 | 70.2 | 64.2 | 47.8 | 64.3 | 42.1 | 58.4 | 43.9 |

**Fully supervised training** In addition to semi-supervised learning, we evaluated *CoLLiS* under a fully supervised setting (Tab. 4). Even with full label availability, all three collaborative models achieve clear improvements over their standalone baselines.

## 4.6. Practical usage with post-training ensemble

In collaborative learning frameworks such as *CoLLiS*, post-training ensemble provides a simple yet effective way to consolidate knowledge from multiple student models. Intuitively, we adopt a straightforward strategy that selects the prediction from the student with the highest confidence:

$$s^* = \arg\max_{s \in 1,...,S}, c(P_s), \quad P_{ensemble} = P_{s^*}. \quad (13)$$

*Table 11.* **Supervised training** results on nuScenes dataset. We do not apply test time augmentation or ensembling during inference.

| Method | FRNet | PolarNet | Cylinder3D |
|---|---|---|---|
| sup. only | 77.7 | 70.4 | 72.1 |
| *CoLLiS* | 78.4 (+0.7) | 73.2 (+2.8) | 75.3 (+3.2) |

As shown in Tab. 12, ensemble predictions achieve higher accuracy than individual models. However, such ensembling is impractical for deployment, as fusing multiple collaborative models introduces additional computational overhead. To transfer this benefit to a more efficient model, we treat the ensemble predictions as high-quality pseudo-labels for offline distillation. Specifically, we distill a lightweight network, FIDNet (Zhao et al., 2021), using the ensemble outputs of three collaboratively trained models. This two-stage extension is particularly suitable for real-time applications with resource-constrained deployment, where lightweight models are preferred and direct collaboration across heterogeneous representations may be impractical due to large performance gaps. In Tab. 13, this strategy yields clear performance gains without modifying the training framework, highlighting the broader applicability of our method beyond direct ensemble inference.

*Table 12.* Evaluation of **post-training ensemble**. As reference, we provide the results of best-performing model from *CoLLiS* evaluated standalone in the first row.

| Method | nuScenes | | SemanticKITTI | |
|---|---|---|---|---|
| | 1% | 10% | 1% | 10% |
| w/o ensemble | 63.2 | 74.2 | 56.0 | 64.3 |
| w/ensemble | 64.5 | 75.5 | 57.6 | 66.4 |

*Table 13.* Results of *CoLLiS* with the two-stage extension. (+Δ) indicates the improvement gained by extending *CoLLiS* with offline distillation.

| Repr. | Method | nuScenes | | SemanticKITTI | |
|---|---|---|---|---|---|
| | | 1% | 10% | 1% | 10% |
| | sup. only | 38.3 | 57.5 | 36.2 | 52.2 |
| | CPS (Chen et al., 2021a) | 40.7 | 60.8 | 36.5 | 52.3 |
| Range | LaserMix (Kong et al., 2023c) | 49.5 | 68.2 | 43.4 | 58.8 |
| | IT2 (Liu et al., 2024) | 56.5 | 71.3 | 51.9 | 60.3 |
| | *CoLLiS* | 57.8 | 70.8 | 50.3 | 61.0 |
| | *CoLLiS* + offline distillation | 60.1 (+2.3) | 73.5 (+2.7) | 53.5 (+3.2) | 63.9 (+2.9) |

## 4.7. Discussion

**Limitations** Despite effectively mitigating confirmation bias, *CoLLiS* still struggles with extremely rare classes. This limitation primarily stems from severe class imbalance. For instance, the *bicycle* class accounts for only 0.01% of the total annotations in the nuScenes dataset. In such cases, annotations are extremely sparse, as illustrated by two examples in Fig. 8. This extreme scarcity makes it inherently difficult for models to accurately learn and predict these isolated points. As a preliminary attempt to address this issue, we explored class-rebalancing strategies such as long-tail class pasting (Xiao et al., 2022). While these techniques yield

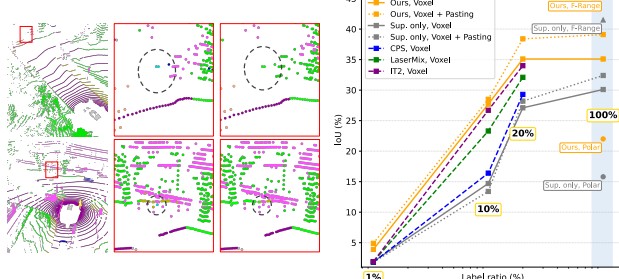

*Figure 8.* Visualization of failure cases. From left to right, columns show the scene overview, ground-truth annotations, and model predictions. Points colored in light blue correspond to the *bicycle* class. The plot on the left reports the IoU for this class.

modest improvements, they remain insufficient to close the performance gap. These results suggest that the bottleneck arises from intrinsic data scarcity rather than from the collaborative learning paradigm itself. Nevertheless, *CoLLiS* consistently improves performance over individual representations compared to prior LiDAR SemiSL approaches.

**Future works** *CoLLiS* still faces challenges with extremely rare long-tail classes due to performance degradation shared across all collaborative models. Future work will explore stronger rebalancing techniques to enhance the generalization ability (Chang et al., 2024; Wei et al., 2021). Another promising direction is the efficient design of contrastive learning across multiple representations. Standard contrastive objectives incur substantial computational overhead, which may limit the scalability of multi-representation training. Moreover, *CoLLiS* does not include multi-modality or temporal cues for training. Integrating collaborative learning with richer supervision sources, e.g. image-LiDAR alignment (Sautier et al., 2022; Liu et al., 2023), language guidance (Kong et al., 2025), geometric priors (Pittner et al., 2024; Yang et al., 2026) and temporal consistency (Lin et al., 2025; Xu et al., 2024; Pittner et al., 2025) may further enhance label-efficient training in real-world autonomous driving applications. We leave them as valuable directions for future explorations.

## 5. Conclusion

In this work, we present *CoLLiS*, a collaborative learning framework for semi-supervised LiDAR semantic segmentation. *CoLLiS* trains multiple networks on different LiDAR representations as coequal students within a single streamlined process. The framework scales multi-representation learning efficiently, mitigates confirmation bias through balanced knowledge transfer, and improves generalization via adaptive data augmentation. We validate the effectiveness of *CoLLiS* through extensive experiments on three public benchmarks under diverse evaluation settings, demonstrating consistent improvements over state-of-the-art methods.

## Impact Statement

This paper presents work whose goal is to advance the field of Machine Learning. There are many potential societal consequences of our work, none of which we feel must be specifically highlighted here.

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

# A. Additional details

## A.1. Dataset

nuScenes (Fong et al., 2022) contains 1,000 driving scenes captured by a 32-beam LiDAR, annotated with 16 semantic classes after merging similar and infrequent classes. SemanticKITTI (Behley et al., 2019) consists of 22 sequences captured with a 64-beam LiDAR and includes 19 semantic classes, with sequences 00–07/09–10 (19,130 scans) for training and 08 (4,071 scans) for validation. ScribbleKITTI (Unal et al., 2022) shares the same point cloud data of SemanticKITTI but replaces full annotations with sparse scribbles (8.06% labeled points for training).

## A.2. Implementation

The initial confidence threshold ($\delta_0$) is set to 0.95 for nuScenes (Fong et al., 2022) and 0.9 for other two datasets. The initial weights of unlabeled loss ($\lambda_0$) are predefined based on scarcity level for nuScenes and SemanticKITTI (Behley et al., 2019): (1, 0.5, 0.5, 0.3) for labeled proportions of (1%, 10%, 20%, 50%), while for ScribbleKITTI (Unal et al., 2022), the weight is fixed at 1 due to its inherently sparse annotations.

In the *multi-representation* setting, We deploy FRNet (Xu et al., 2023), Cylinder3D (Zhu et al., 2021) and PolarNet (Zhang et al., 2020). For FRNet, the 2D representation shape is set to $64 \times 512$ for SemanticKITTI (Behley et al., 2019) and ScribbleKITTI (Unal et al., 2022), and $32 \times 480$ for nuScenes (Fong et al., 2022). For PolarNet and Cylinder3D, the grid size is reduced to $240 \times 180 \times 20$, following the settings in prior works (Kong et al., 2023c; Liu et al., 2024) for fair comparison.

In the *dual-representation* setting, we follow the same configuration of IT2 (Liu et al., 2024), where a range-view network FIDNet (Zhao et al., 2021) and Cylinder3D (Zhu et al., 2021) are incorporated in the framework.

## A.3. Training details

We use a batch size of 14 for both labeled and unlabeled data (effective batch size is then 28 after mixing) for nuScenes (Fong et al., 2022) dataset. The learning rate is set to $6e^{-3}$, and the maximum number of epochs is set to 100. For SemanticKITTI (Behley et al., 2019) and ScribbleKITTI (Unal et al., 2022), the batch size is reduced to 8 and learning rate is $8e^{-3}$. The maximum number of epochs is 95. For all experiments in the work, we use the AdamW (Loshchilov & Hutter, 2017) optimizer with a weight decay of 0.0001 and a OneCycleLR scheduler (Smith & Topin, 2018), and employ a single NVIDIA H200 GPU for training.

## A.4. Evaluation metrics

The performance is measured with average Intersection-over-Union (mIoU).

## A.5. Mixing strategies

To enhance LiDAR point cloud augmentation, we integrate three geometrically complementary mixing strategies, each tailored to exploit distinct spatial properties of LiDAR data. In Fig. 9, we visualize the point clouds mixed by different strategies. Notably, the visualization shows that mixed point clouds have distinct difference in geometry. Below, we provide technical details of every mixing strategy:

1. LaserMix (Kong et al., 2023c) partitions two scenes along elevation angle (vertical sweep axis) and interleaves their sectors while keeping the ring-like geometry of the scene. The mixing generally follows the inherent scan pattern of LiDAR sensor.

2. PolarMix (Xiao et al., 2022) operates in the polar coordinate space, splitting point clouds radially and horizontally into different several regions. Mixed scenes retain intact local object geometry by constraining swaps to entire polar regions.

3. Sub-cloud Shuffling (Yang & Condurache, 2026) downsamples two point clouds into two sub-clouds, respectively, then randomly interleaves them. this preserves local coherence and scene integrity while fusing semantic contexts from both scenes

Overall, these three mixing strategies augment point clouds from different geometric perspectives. By combining them, we harness complementary spatial transformations that capture both local and global structures, enhancing the diversity of geometric patterns. This synergy increases the generalization capability of data augmentation within our framework, leading to more robust and adaptable semi-supervised LiDAR semantic segmentation.

# B. Additional experiments

## B.1. Heterogeneous *CoLLiS*

Besides development in using different representations for LiDAR semantic segmentation. parallel advancements have occurred in architectural design. Driven by the success of Transformers (Vaswani et al., 2017) in vision tasks (Dosovitskiy et al., 2021), recent works increasingly replace conventional CNNs with Transformer-based backbones for LiDAR segmentation (Ando et al., 2023; Kong et al., 2023a; Lai et al., 2023; Wu et al., 2024), leveraging their ability to model long-range dependencies in sparse 3D data.

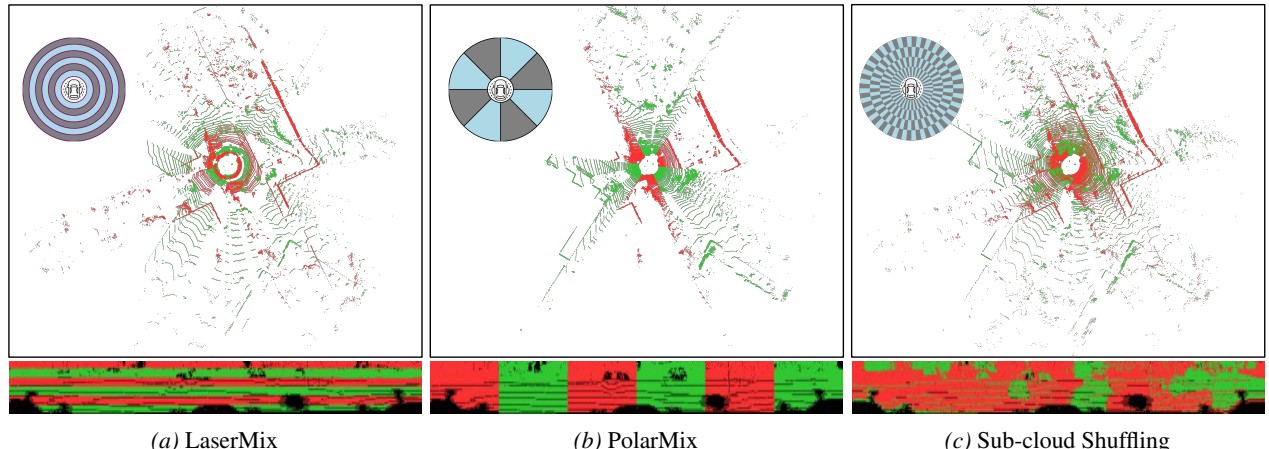

*(a) LaserMix*       *(b) PolarMix*       *(c) Sub-cloud Shuffling*

*Figure 9.* Examples of different mixing strategies using two LiDAR point clouds (distinguished by green and red). Mixed point clouds are visualized in bird's-eye view (top) and range view (bottom).

To additionally evaluate the impact of architectural diversity independently of representational differences, we further test *CoLLiS* in a setting where the same LiDAR representation is processed by two networks with heterogeneous architectures. In this scenario, we integrate two range-view methods, FIDNet (Zhao et al., 2021) and RangeViT (Ando et al., 2023). The shape of range-view image is set to $32 \times 1920$ for nuScenes and $64 \times 2048$ for ScribbleKITTI.

Tab. 14 demonstrates that *CoLLiS* achieves significant improvements in semi-supervised scenarios even with a single representation by integrating heterogeneous architectures (e.g., CNNs and ViTs). For instance, with 1% labeled data on ScribbleKITTI (Unal et al., 2022), combined networks outperforms LaserMix (Kong et al., 2023c) by +2.5% mIoU while trained on the same range-view inputs. In other settings, collaboration of heterogeneous networks exhibits the competitive performance as well. This underscores that architectural diversity drives robust representation learning in label-scarce regimes.

## C. Additional results

### C.1. Detailed results

We provide the detailed results of class-wise IoU in Tab. 15, Tab. 16 and Tab. 17.

### C.2. Qualitative results

We provide additional qualitative results in Fig. 10.

*Table 14.* Evaluation of **FIDNet** (Zhao et al., 2021) with **heterogeneous** *CoLLiS*. The performance is compared with other single-representation approaches. *: We re-implement the IT2 (Liu et al., 2024) framework with heterogeneous architectures for fair comparison. The best results are highlighted in **bold**.

| Method | nuScenes (Fong et al., 2022) | | | ScribbleKITTI (Unal et al., 2022) | | |
|---|---|---|---|---|---|---|
| | 1% | 10% | 20% | 1% | 10% | 20% |
| sup. only | 38.3 | 57.5 | 62.7 | 33.1 | 47.7 | 49.9 |
| CPS (Chen et al., 2021b) | 40.7 | 60.8 | 64.9 | 33.7 | 50.0 | 52.8 |
| LaserMix (Kong et al., 2023c) | 49.5 | 68.2 | **70.6** | 38.3 | 54.4 | 55.6 |
| *IT2-Hete (Liu et al., 2024) | 50.1 | **68.3** | 69.9 | 39.5 | 53.4 | **57.8** |
| *CoLLiS-Hete* | **50.3** | 67.7 | 69.6 | **40.8** | **55.1** | 56.2 |

*Table 15.* The class-wise IoU results in nuScenes (Fong et al., 2022) dataset among different partition protocol. The mIoU results are highlighted in red. 100% label proportion denotes the fully supervised training results.

| Repr. | prop. | mIoU | barr | bicy | bus | car | const | moto | ped | cone | trail | truck | driv | othe | walk | terr | manm | veg |
|---|---|---|---|---|---|---|---|---|---|---|---|---|---|---|---|---|---|---|
| F-Range | 1% | 63.2 | 66.5 | 3.1 | 78.4 | 86.9 | 14.6 | 66.3 | 72.3 | 50.2 | 35.8 | 60.0 | 95.3 | 65.1 | 69.5 | 73.9 | 86.2 | 86.2 |
| | 10% | 74.2 | 76.9 | 0.8 | 92.6 | 85.7 | 54.2 | 84.7 | 73.1 | 66.2 | 68.7 | 84.0 | 96.7 | 75.9 | 75.9 | 77.0 | 88.1 | 86.7 |
| | 20% | 74.8 | 77.9 | 0.7 | 93.4 | 91.7 | 55.2 | 85.4 | 76.2 | 67.1 | 67.6 | 83.6 | 96.7 | 74.5 | 75.5 | 76.6 | 88.3 | 86.6 |
| | 50% | 75.8 | 77.6 | 35.1 | 92.7 | 91.1 | 50.4 | 82.5 | 73.3 | 66.1 | 65.2 | 82.2 | 96.5 | 73.1 | 74.5 | 76.3 | 88.8 | 87.4 |
| | 100% | 78.4 | 78.3 | 45.2 | 92.6 | 91.7 | 58.1 | 84.4 | 77.9 | 67.8 | 71.1 | 83.5 | 96.7 | 77.3 | 76.1 | 77.1 | 89.0 | 87.5 |
| Polar | 1% | 57.9 | 59.5 | 2.1 | 78.4 | 83.8 | 7.8 | 61.4 | 51.7 | 41.0 | 31.7 | 58.8 | 93.1 | 60.3 | 63.8 | 69.2 | 82.7 | 81.8 |
| | 10% | 68.4 | 71.4 | 11.3 | 90.0 | 87.6 | 39.1 | 69.7 | 58.5 | 51.3 | 63.6 | 77.9 | 94.9 | 69.1 | 70.4 | 71.8 | 85.2 | 82.9 |
| | 20% | 68.6 | 72.8 | 1.6 | 90.9 | 88.2 | 43.0 | 73.1 | 57.8 | 50.4 | 64.6 | 78.6 | 94.9 | 70.6 | 70.9 | 72.1 | 85.4 | 82.8 |
| | 50% | 70.8 | 72.3 | 15.9 | 92.1 | 88.4 | 47.5 | 75.4 | 58.4 | 52.4 | 67.6 | 79.6 | 95.0 | 73.1 | 71.7 | 72.3 | 85.3 | 85.2 |
| | 100% | 73.2 | 75.4 | 22.0 | 93.3 | 89.7 | 51.5 | 76.3 | 60.2 | 62.2 | 68.3 | 79.2 | 95.7 | 73.8 | 73.5 | 75.4 | 88.4 | 86.7 |
| Voxel | 1% | 61.1 | 64.2 | 3.9 | 77.2 | 85.4 | 18.8 | 63.0 | 63.7 | 47.4 | 33.8 | 58.5 | 94.0 | 61.3 | 65.2 | 71.0 | 85.3 | 85.0 |
| | 10% | 72.9 | 73.9 | 27.7 | 91.4 | 89.9 | 46.4 | 78.6 | 68.3 | 60.3 | 63.1 | 80.4 | 95.4 | 69.8 | 71.4 | 73.6 | 87.9 | 86.3 |
| | 20% | 73.4 | 74.5 | 35.4 | 92.7 | 90.6 | 45.8 | 80.5 | 69.0 | 60.2 | 67.5 | 81.6 | 95.6 | 72.1 | 72.9 | 75.0 | 87.9 | 86.1 |
| | 50% | 74.5 | 75.8 | 30.1 | 93.0 | 90.8 | 51.0 | 80.1 | 71.7 | 62.1 | 69.7 | 83.4 | 95.8 | 74.7 | 73.8 | 75.3 | 87.9 | 86.3 |
| | 100% | 75.3 | 75.7 | 35.1 | 92.7 | 90.7 | 49.7 | 81.5 | 70.7 | 62.2 | 69.4 | 83.0 | 95.6 | 74.4 | 73.0 | 75.1 | 88.4 | 86.7 |

*Table 16.* The class-wise IoU results in SemanticKITTI (Behley et al., 2019) dataset on different label proportions. The mIoU results are highlighted in red.

| Repr. | prop. | mIoU | car | bicy | moto | truck | o.veh | ped | b.cyc | m.cyc | road | park | walk | o.gro | build | fence | veg | trunk | terr | pole | sign |
|---|---|---|---|---|---|---|---|---|---|---|---|---|---|---|---|---|---|---|---|---|---|
| F-Range | 1% | 56.0 | 91.7 | 16.8 | 48.0 | 66.0 | 50.7 | 61.8 | 68.2 | 0.0 | 90.7 | 50.9 | 73.6 | 0.9 | 85.2 | 42.7 | 86.9 | 44.9 | 73.4 | 60.2 | 41.9 |
| | 10% | 64.3 | 95.7 | 19.1 | 70.1 | 87.3 | 58.8 | 73.1 | 84.8 | 0.0 | 96.0 | 60.5 | 84.0 | 2.3 | 89.6 | 66.2 | 86.5 | 65.8 | 71.9 | 62.4 | 47.4 |
| | 20% | 64.9 | 95.5 | 36.3 | 59.5 | 91.7 | 58.9 | 71.4 | 84.0 | 0.0 | 95.8 | 58.7 | 84.2 | 6.7 | 90.9 | 68.8 | 86.5 | 65.5 | 72.0 | 63.3 | 42.7 |
| | 50% | 66.2 | 95.2 | 47.9 | 71.8 | 92.0 | 51.4 | 73.5 | 81.2 | 0.0 | 96.0 | 61.1 | 83.9 | 12.9 | 89.9 | 63.9 | 87.3 | 65.8 | 73.5 | 63.0 | 47.5 |
| Polar | 1% | 46.8 | 91.2 | 15.5 | 21.1 | 24.6 | 0.3 | 36.0 | 62.4 | 0.0 | 90.8 | 35.7 | 75.5 | 1.1 | 80.3 | 53.2 | 79.8 | 56.7 | 70.8 | 62.1 | 32.2 |
| | 10% | 53.3 | 90.5 | 20.7 | 41.8 | 78.8 | 28.2 | 31.1 | 77.1 | 2.2 | 90.9 | 51.2 | 75.1 | 0.3 | 87.0 | 40.3 | 83.3 | 53.5 | 67.9 | 55.8 | 36.7 |
| | 20% | 54.0 | 91.6 | 23.7 | 45.7 | 75.3 | 29.9 | 35.7 | 75.3 | 0.0 | 91.0 | 52.9 | 74.9 | 0.2 | 88.1 | 43.5 | 84.0 | 52.4 | 70.7 | 54.4 | 36.3 |
| | 50% | 55.5 | 94.1 | 25.7 | 47.3 | 77.0 | 28.8 | 39.4 | 77.0 | 0.0 | 90.7 | 52.2 | 74.5 | 0.5 | 87.5 | 51.9 | 85.7 | 51.4 | 67.1 | 65.2 | 37.9 |
| Voxel | 1% | 53.2 | 91.2 | 21.9 | 43.4 | 66.1 | 29.5 | 32.6 | 74.1 | 0.6 | 90.9 | 49.9 | 74.8 | 0.0 | 87.7 | 47.6 | 84.2 | 51.3 | 70.8 | 53.3 | 40.8 |
| | 10% | 63.1 | 94.0 | 38.2 | 59.3 | 82.4 | 52.0 | 72.2 | 81.0 | 0.9 | 92.5 | 51.8 | 78.4 | 7.8 | 91.1 | 61.8 | 86.6 | 67.8 | 71.4 | 63.4 | 45.8 |
| | 20% | 63.6 | 93.4 | 46.1 | 68.0 | 87.8 | 35.7 | 72.8 | 87.0 | 9.2 | 92.6 | 45.8 | 78.2 | 3.3 | 90.4 | 58.7 | 87.1 | 65.5 | 72.8 | 63.6 | 49.6 |
| | 50% | 64.0 | 94.2 | 50.3 | 67.7 | 89.8 | 43.2 | 71.2 | 88.1 | 4.2 | 92.9 | 49.0 | 78.5 | 0.4 | 90.6 | 60.0 | 86.5 | 67.1 | 71.0 | 63.6 | 48.1 |

*Table 17.* The class-wise IoU results in ScribbleKITTI (Unal et al., 2022) dataset among different partition protocol. The mIoU results are highlighted in red.

| Repr. | prop. | mIoU | car | bicy | moto | truck | o.veh | ped | b.cyc | m.cyc | road | park | walk | o.gro | build | fence | veg | trunk | terr | pole | sign |
|---|---|---|---|---|---|---|---|---|---|---|---|---|---|---|---|---|---|---|---|---|---|
| F-Range | 1% | 47.6 | 90.3 | 30.8 | 27.4 | 27.5 | 4.4 | 37.1 | 66.3 | 0.0 | 83.7 | 36.3 | 71.9 | 3.3 | 87.0 | 39.9 | 80.3 | 58.3 | 66.1 | 56.1 | 38.2 |
| | 10% | 59.9 | 94.4 | 28.6 | 61.9 | 83.6 | 42.7 | 64.4 | 80.4 | 0.0 | 87.1 | 46.7 | 75.0 | 0.6 | 89.8 | 54.4 | 84.7 | 65.7 | 68.9 | 63.5 | 46.8 |
| | 20% | 60.1 | 94.6 | 30.6 | 60.9 | 92.2 | 46.4 | 64.2 | 83.2 | 0.0 | 86.3 | 46.1 | 74.1 | 0.5 | 88.7 | 49.9 | 83.5 | 64.4 | 65.9 | 63.4 | 47.9 |
| | 50% | 60.7 | 95.1 | 31.1 | 58.6 | 88.4 | 50.6 | 63.3 | 85.4 | 0.0 | 86.6 | 43.1 | 74.3 | 0.7 | 89.6 | 52.3 | 84.9 | 66.2 | 69.7 | 63.8 | 48.7 |
| Polar | 1% | 42.0 | 89.2 | 23.2 | 18.0 | 18.9 | 3.1 | 20.9 | 61.6 | 0.3 | 82.0 | 23.5 | 66.8 | 0.7 | 86.9 | 27.4 | 77.9 | 49.5 | 61.6 | 52.4 | 33.4 |
| | 10% | 51.1 | 91.1 | 16.1 | 49.1 | 52.6 | 34.0 | 34.2 | 81.1 | 0.0 | 84.7 | 40.9 | 69.4 | 0.1 | 87.4 | 33.9 | 81.8 | 55.6 | 68.1 | 55.5 | 36.3 |
| | 20% | 51.4 | 91.1 | 15.5 | 48.7 | 57.8 | 34.2 | 33.5 | 81.0 | 0.0 | 84.7 | 40.8 | 69.4 | 0.1 | 87.5 | 33.9 | 81.7 | 55.4 | 68.0 | 55.4 | 35.8 |
| | 50% | 51.7 | 91.3 | 18.2 | 47.4 | 72.5 | 32.0 | 36.3 | 73.8 | 0.4 | 83.7 | 42.4 | 68.5 | 0.1 | 86.4 | 36.4 | 81.7 | 54.3 | 65.8 | 55.4 | 35.1 |
| Voxel | 1% | 47.6 | 90.4 | 30.8 | 12.2 | 20.9 | 2.2 | 41.1 | 71.1 | 0.2 | 82.2 | 25.4 | 68.4 | 0.7 | 88.2 | 31.9 | 80.6 | 63.2 | 65.3 | 61.5 | 39.7 |
| | 10% | 58.6 | 93.3 | 29.6 | 56.1 | 68.3 | 40.2 | 61.6 | 83.5 | 8.9 | 85.9 | 39.8 | 72.4 | 1.1 | 89.4 | 46.2 | 86.0 | 66.6 | 71.9 | 64.4 | 49.2 |
| | 20% | 58.8 | 93.3 | 31.3 | 59.8 | 79.4 | 38.7 | 64.4 | 82.3 | 0.0 | 85.0 | 35.4 | 72.1 | 3.6 | 89.2 | 43.8 | 85.3 | 68.1 | 71.0 | 64.0 | 49.5 |
| | 50% | 59.0 | 93.5 | 29.9 | 60.3 | 84.4 | 41.3 | 64.5 | 84.7 | 6.3 | 84.3 | 36.0 | 71.2 | 0.7 | 89.1 | 42.2 | 84.4 | 67.3 | 68.7 | 63.0 | 49.7 |

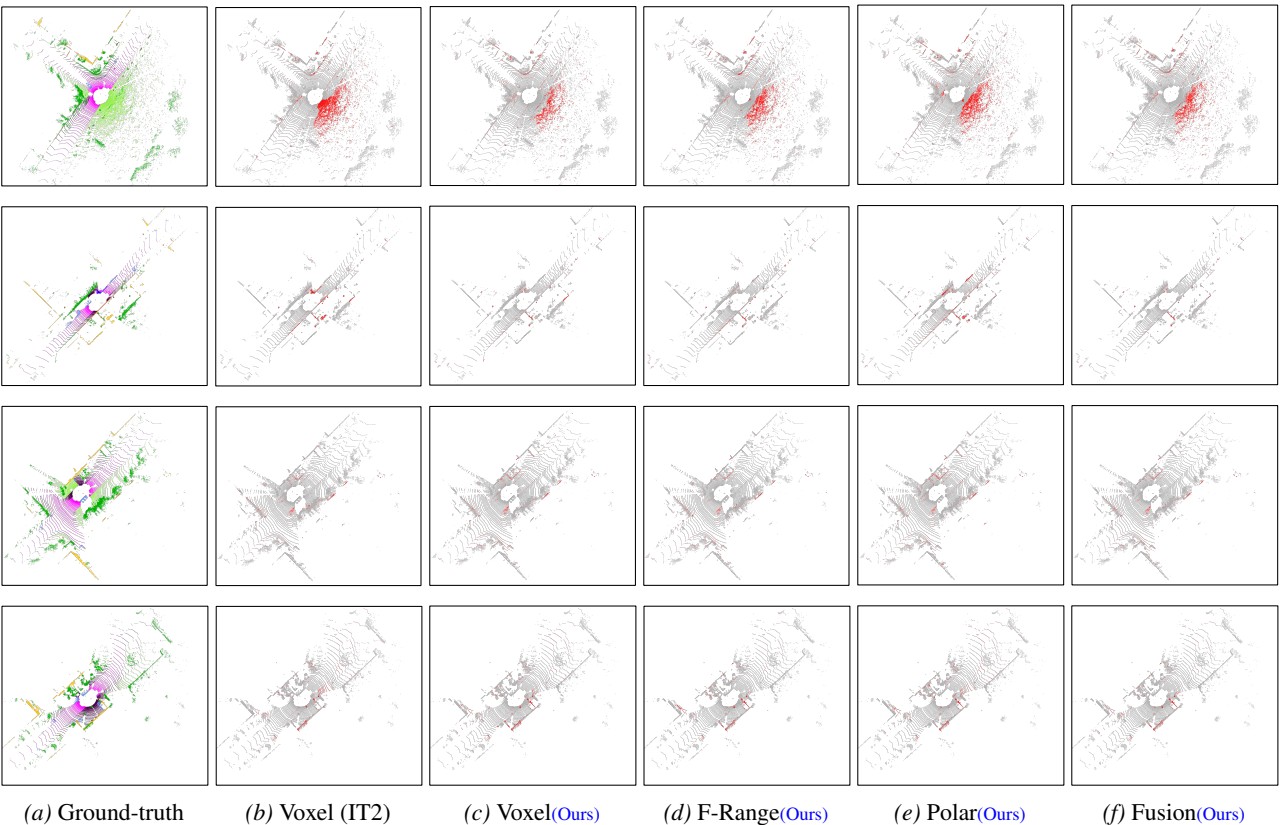

*(a)* Ground-truth   *(b)* Voxel (IT2)   *(c)* Voxel(Ours)   *(d)* F-Range(Ours)   *(e)* Polar(Ours)   *(f)* Fusion(Ours)

*Figure 10.* **Qualitative results** on SemanticKITTI (Behley et al., 2019). All models are trained under the 10% label protocol. We use Hard Confidence Voting (HCV) to fuse students' outputs. Ground-truth labels are color-coded based on class categories. Incorrect predictions are shown in red, while correct predictions are shown in gray.

