# OpenReview forum: "Collaborative Learning for Semi-Supervised LiDAR Semantic Segmentation"
_ICML.cc/2026/Conference — ICML 2026 regular_

### Official Review · Reviewer_h8ba · 2026-03-06

**Soundness:** 3
**Presentation:** 3
**Significance:** 3
**Originality:** 3
**Overall Recommendation:** 5
**Confidence:** 3

**Summary:**

This paper proposes CoLLiS, a semi-supervised learning framework designed to improve LiDAR-based 3D semantic segmentation under extremely limited label settings. The authors point out that existing approaches are vulnerable to confirmation bias due to their reliance on a single pseudo-label source, and introduce a collaborative learning structure across multiple LiDAR representations to mitigate this issue.
Specifically, models based on different representations are treated as equal-status students and perform mutual distillation. The framework further incorporates confidence-based pseudo-label selection and dynamic data augmentation to reduce error propagation.
Experiments on major benchmarks demonstrate consistent performance improvements over prior methods, particularly in low-label regimes, and empirically validate the effectiveness of reducing confirmation bias.

**Compliance With Llm Reviewing Policy:**

Affirmed.

**Final Justification:**

The authors’ responses were sufficiently convincing, and thus I believe the original score should be maintained.

**Key Questions For Authors:**

- In Fig. 5(a), the trends of polar→voxel and polar→frustum appear similar, and frustum→voxel and frustum→polar also show comparable behavior, whereas voxel→polar and voxel→frustum exhibit different patterns. Is there a structural reason for this asymmetry? For example, could it be related to differences in information density, discretization granularity, or inductive bias across representations?

- In the training efficiency analysis, the multi-representation approach is compared with IT2, but a direct comparison in terms of latency and memory consumption against single-representation methods (without voxel-based branches) is not provided. Could the authors quantify the actual computational and memory overhead relative to a single-representation baseline?

- After offline distillation, the lightweight model appears to outperform the baseline CoLLiS in Table 10. Does Table 10’s CoLLiS refer to a single-model performance, while the distillation stage uses the most confident prediction among collaborative models as the teacher? In other words, is the observed gain primarily due to an ensemble effect, or does the distillation process itself introduce additional regularization benefits? A clearer explanation would be helpful.

- In a multi-representation collaborative framework, if one representation performs significantly worse than the others, could it act as a noise source and negatively influence the collaboration process? How robust is the framework to imbalanced representation quality?

- The paper includes OOD robustness evaluations. Is there potential to extend the proposed framework toward domain adaptation (DA) settings? For instance, could the collaborative mechanism be adapted to scenarios with explicit source–target distribution gaps? A brief discussion on this possible extension would be valuable.

**Limitations:**

yes

**Strengths And Weaknesses:**

# Soundness
**Strengths**
- The paper clearly identifies the vulnerability of existing semi-supervised LiDAR segmentation methods to confirmation bias due to reliance on a single pseudo-label source, and proposes an adaptive pseudo-labeling strategy that combines AR (Absolute Reliability) and RR (Relative Reliability). This design appears methodologically sound and well motivated.
- The integration of stabilization mechanisms—such as threshold adjustment, adaptive weighting, confidence regularization, and CDA—demonstrates a systematic consideration of instability issues inherent in semi-supervised learning.
- The experimental setup is comprehensive, covering multiple datasets, varying label ratios, scalability analysis, and OOD settings, providing sufficient empirical support for the effectiveness of the proposed approach.

**Weaknesses**
- In evaluating pseudo-label quality, the paper relies on retention rate and certainty metrics. Including more direct quality measures—such as precision and recall of pseudo-labels—would strengthen the empirical analysis and improve persuasiveness.

# Presentation
**Strengths**
- The paper first explains the limitations of conventional two-step pseudo-labeling frameworks and then naturally motivates the transition to a single-step collaborative structure.
- Key components such as AR–RR-based adaptive pseudo-labeling and CDA are introduced in a step-by-step manner, making the overall pipeline relatively clear and easy to follow.
- The experimental section is well organized, presenting comparative evaluations, ablation studies, and extended analyses in a structured way, which facilitates interpretation of the results.

**Weaknesses**
- Some tables and figures could be improved in terms of readability. In particular, the right-hand side of Fig. 8 would benefit from adjustments in font size and visual contrast to enhance clarity.

# Significance
**Strengths**
- The paper addresses the practically important problem of low-label LiDAR semantic segmentation and proposes a structural approach to mitigate confirmation bias through multi-representation collaboration.
- By extending single pseudo-label–based semi-supervised learning toward a multi-representation collaborative framework, and demonstrating consistent improvements in low-label regimes, the work offers meaningful practical contributions.

**Weaknesses**
- The proposed approach assumes an environment where multiple representations can be trained in parallel, which may limit applicability under resource-constrained settings.
- Since the primary contribution lies in performance improvement and stabilization, rather than introducing a fundamentally new problem formulation or paradigm shift, the overall impact may be considered incremental rather than transformative.

# Originality
**Strengths**
- In the context of semi-supervised LiDAR segmentation, the paper demonstrates applied originality by structurally integrating AR–RR-based adaptive distillation with multiple stabilization strategies to mitigate confirmation bias.
- Beyond simple mutual distillation, the combination of filtering, threshold adaptation, adaptive weighting, and regularization differentiates the framework at the design level.

**Weaknesses**
- The core idea of improving pseudo-label quality through multi-representation collaboration can be viewed as an extension of prior DML or dual-representation approaches, rather than the introduction of an entirely new learning principle.
- Overall, the contribution is closer to a well-engineered and carefully adapted extension of existing concepts to the LiDAR semi-supervised setting, rather than a fundamentally new methodological breakthrough.

---

> ### Author Rebuttal · Authors · 2026-03-30
>
> We truly appreciate the rveiewer for the careful and constructive feedback. We are particularly encouraged by the reviewer's recoginition on well-motivated design, comprehensive experiments and practical advantages of our method. We address each concern below and will include all discussion and additional results in the revision.
> ### Q1: Asymmetry in Fig. 5(a)
> This is an insightful observation. The asymmetry can be explained by two factors: architectural similarity and geometric compatibility.
>
> FRNet and PolarNet are both 2D convolution-based networks that converge faster than Cylinder3D, which relies on sparse 3D convolutions. This explains why frustum→X and polar→X each show consistent retention rates regardless of the target. The asymmetry unique to voxel arises from its differing compatibility with each target. Voxel→polar increases smoothly, as Cylinder3D and PolarNet share polar coordinate geometric priors, enabling natural pseudo-label exchange as the voxel model matures. In contrast, voxel↔frustum is more dynamic, as these two differ in both architectural design (3D sparse conv vs. 2D conv) and geometric prior (cylindrical vs. spherical projection). Additionally, voxel discretization introduces spatial quantization that reduces prediction sharpness in sparse regions compared to frustum-range, which preserves raw scan density.
>
> Importantly, this behaviour is **expected and not a sign of instability**. Voxel and frustum carry the most unique and non-redundant information about each other. This complementarity is precisely the primary driver of generalization gains in our collaborative framework. The eventual high retention rate between the two confirms that the framework is self-correcting and stable.
> ### Q2: Computational/memory overhead
> Please see our response to Q1 of Reviewer ZsAL.
> ### Q3: Explanation on offline distillation gain
> We apologize for the lack of clarity. To clarify: the CoLLiS entry in Tab. 10 refers to the standalone performance of FIDNet trained within the collaborative framework alongside voxel and polar representations, not the ensemble. The offline distillation stage then uses the ensemble prediction as **hard pseudo-labels** to further train FIDNet.
>
> In our implementation, the gain is mianly attributed to the ensemble teacher being stronger than any individual student, rather than additional regularization from distillation itself. To verify this, we report the performance of the best single collaborative student ("w/o ensemble") against the ensemble predictions below:
> ||nuScenes 1%|nuScenes 20%|SemKITTI 1%|SemKITTI 20%|
> |-|-|-|-|-|
> |w/o ensemble|63.2|74.2|56.0|64.3|
> |w/ ensemble|64.5|75.5|57.6|66.4|
>
> More broadly, the two-stage extension is motivated by a practical scenario: when a lightweight or underperforming network cannot participate in collaboration on equal footing with stronger representations, it can still benefit from the distilled knowledge of a strong collaborative ensemble as a post-hoc step. This makes CoLLiS applicable to **resource-constrained deployment** settings where only a single lightweight model is feasible at inference time.
> ### Q4: Robustness to imbalanced representation quality
> Such imbalance is naturally handled by our collaborative mechanism. As shown in Fig. 5(a), the pseudo-label retention rate of the polar representation is averagely lower than that of the other two. This is consistent with Tab. 1, where polar representations generally underperform. Our RR mechanism and adaptive distillation weights down-weight pseudo-labels from weaker students, preventing noise amplification and ensuring that stronger representations are not misled by less reliable sources.
>
> Notably, the scalability experiments in Tab.4 also provides an empirical evidence. When the polar representation is added as a third collaborator, FRNet improves from 62.5 to 63.2 and Cylinder3D from 59.2 to 61.1. Rather than introducing noise, weaker models contribute complementary pseudo-labels in regions where it is locally confident, while being appropriately moderated. This demonstrates that CoLLiS is robust to imbalanced representation quality in both directions: weaker representations benefit substantially from collaboration, while stronger representations are marginally improved by the inclusion of weaker ones. We believe such dynamic is the key advantage of our reliability-aware collaborative design.
> ### Q5: Extension to domain adaptation (DA)
> We believe the collaborative mechanism is naturally compatible with domain adaptation settings, as leveraging diverse representations can improve resilience to distribution shift. A rigorous formulation of CoLLiS for explicit source-target adaptation is an interesting direction that we leave for future work.
> ### W: Fig. 8 readability and pseudo-label precision/recall
> We thank the reviewer for the valuable suggestion. We will improve readablity of Fig.8. We will also include pseudo-label precision and recall metrics in the revised manuscript.

---

> > ### Author Rebuttal · Reviewer_h8ba · 2026-04-01
> >
> > The authors’ responses have resolved my concerns, and I believe that my original score was appropriate.

---

> > > ### Author Response · Authors · 2026-04-01
> > >
> > > We sincerely thank the reviewer for engaging with our rebuttal. We are glad that our responses have addressed the concerns. The reviewer’s suggestions are valuable and will significantly strengthen the paper, and we will incorporate them in the final version.

---

### Official Review · Reviewer_ZsAL · 2026-03-11

**Soundness:** 3
**Presentation:** 2
**Significance:** 3
**Originality:** 3
**Overall Recommendation:** 4
**Confidence:** 2

**Summary:**

The submission introduces CoLLiS, a novel framework designed to effectively train LiDAR semantic segmentation models with limited annotated data. To address the vulnerability of current semi-supervised learning (SemiSL) frameworks to "confirmation bias"—which arises from a reliance on single-source pseudo-labels in decoupled, two-step paradigms—CoLLiS treats multiple distinct LiDAR representations (such as range views, voxel grids, and polar images) as coequal "students" and trains them collaboratively in a single forward step.

**Compliance With Llm Reviewing Policy:**

Affirmed.

**Final Justification:**

The response reinforced my prior assessment of the manuscript.

**Key Questions For Authors:**

1.
CoLLiS involves the simultaneous training of multiple representation heads. Could you provide a detailed comparison of the total training time and peak GPU memory consumption of CoLLiS compared to a single-representation baseline and a standard two-step distillation method?

2.
The CDA mechanism relies on inter-student agreement to determine the intensity of augmentation. In scenarios where all student models are equally confident but converge on the same incorrect prediction, how does the framework prevent the reinforcement of a collective "confirmation bias"?

3.
The experiments primarily use Voxel and Range representations. How sensitive is the performance to the architectural diversity of the students? For example, if two students use highly similar architectures or representations, does the "Relative Reliability" (RR) metric lose its effectiveness?

**Limitations:**

The CoLLiS framework presents a technically sound and innovative approach to semi-supervised LiDAR semantic segmentation by replacing the conventional two-step teacher-student paradigm with a single-step "coequal students" collaborative learning architecture. Its core strength lies in its ability to effectively mitigate confirmation bias through consensus-driven knowledge transfer among diverse LiDAR representations, a claim robustly supported by results across major benchmarks, particularly in low-label scenarios. While the simultaneous training of multiple representation heads introduces non-trivial computational overhead and the framework's current scope remains specialized to LiDAR-specific geometries, the introduction of adaptive metrics like Relative Reliability (RR) and Consensus-Driven Augmentation (CDA) represents a significant contribution that enhances the practical utility of 3D perception systems under limited supervision.

**Strengths And Weaknesses:**

The paper is technically solid and addresses the critical "confirmation bias" issue in semi-supervised LiDAR segmentation through a well-reasoned collaborative learning paradigm. The transition from a decoupled two-step distillation to a single-step coequal student framework is logically sound. The authors provide comprehensive empirical validation across three major benchmarks, demonstrating consistent performance gains over state-of-the-art methods like LaserMix and PVKD, particularly in low-label regimes (1% and 5%). The introduction of "Relative Reliability" (RR) to handle contradictory supervision is a mathematically grounded approach to ensuring label quality. While the single-step design improves training flow, the simultaneous optimization of multiple representation heads (Voxel, Range, Point) inevitably increases the per-iteration computational cost and GPU memory footprint. The paper would benefit from a more detailed quantitative analysis of the trade-off between this increased training overhead and the final performance gain.

The manuscript is exceptionally well-structured and follows a clear narrative arc. The motivation is vividly illustrated. The authors do an excellent job of positioning their work within the context of existing literature, clearly articulating the limitations of prior "single-source" pseudo-labeling. The inclusion of qualitative results and detailed hyperparameter settings facilitates reproducibility. There are minor typographical errors and some dense mathematical notations in the loss function descriptions that could be slightly more streamlined for a general ML audience.

This work addresses a high-impact problem: the prohibitive cost of 3D point cloud annotation for autonomous driving. By enabling more effective learning from unlabeled data, CoLLiS offers immediate practical utility for scaling perception systems. The framework’s versatility—demonstrated by its ability to integrate various architectures like PVv2 and PVv3—suggests it could become a standard paradigm for multi-representation 3D learning. The scope is currently specialized for LiDAR data. While significant within the 3D vision community, the cross-representation mapping techniques (e.g., range-to-voxel) are domain-specific, which might limit its immediate application to other semi-supervised tasks like 2D image processing without significant adaptation.

The originality of CoLLiS lies in its creative shift from a "Teacher-Student" hierarchy to a "Collaborative Student" ecosystem. The "Consensus-Driven Augmentation" (CDA) is a novel concept that dynamically adjusts the mixing ratio based on inter-student agreement, turning the consistency of different views into a supervisory signal. The use of point-wise reliability counts (RR) rather than simple confidence averaging provides a fresh perspective on resolving multi-view conflicts. Some individual components, such as the underlying mixing strategies (LaserMix/PolarMix), are borrowed from existing literature. However, the novel combination and the dynamic logic governing their interaction represent a distinct and valuable contribution to the field.

---

> ### Author Rebuttal · Authors · 2026-03-30
>
> We sincerely thank the reviewer for the detailed and encouraging assessment. We especially appreciate the reviewer’s recognition that CoLLiS is a novel framework for LiDAR SemiSL and well-reasoned way for addressing the critical issue of confirmation bias. We are also grateful for the positive comments on the broader significance and solid execution of the work. We address each concern below and will include all discussion and additional results in the revision.
> ### Q1: Total Training time and GPU memory comparison
> We provide memory consumption in Tab. 3, where LaserMix serves as the single-representation baseline with standard two-step training. For total training time on a single NVIDIA H200 GPU, we report the following:
> |Method|Training Time|
> |-|-|
> |LaserMix|15.4 hours|
> |IT2|50.8 hours|
> |CoLLiS| 25.6 hours|
>
> CoLLiS requires approximately 60% more training time than LaserMix, which is expected given that three representations are trained simultaneously. However, compared to IT2, a two-representation method with a two-step design, CoLLiS is 2× faster while training an additional representation. This efficiency advantage stems directly from our streamlined single-step design, which avoids additional time-consuming components such as contrastive loss computation required by IT2. We acknowledge that training time accumulates as more representations are added. As future work, we would like to explore how the framework could be further compressed for an increasing number of representations. One promising direction is to incorporate a mixture-of-experts (MoE) mechanism that allows students to be selectively activated during collaboration, so that not all students need to participate at every training iteration.
> ### Q2: Preventing collective confirmation bias when all students converge on the same wrong predictions
> This is an insightful question. CoLLiS mitigates the risk of all students collectively converging on the same incorrect prediction through several mechanisms. First, confidence regularization loss (Eq. 11) discourages overconfident errors by smoothing predictions, softening the collective error signal. Second, deploying multiple representations with fundamentally different inductive bias makes it structurally unlikely for all students to fail similarly with high confidence. Third, CDA uses a fixed step size to update the mixing probability, which prevents overly reactive updates in such corner cases. As shown in Fig. 7 (right), too small step size leads to performance degradation.
>
> That said, we acknowledge that these edge cases are non-negligible. As discussed in Sec.4.7, all representations can misclassify extremely rare classes that correspond to very few isolated points in the scene (Fig.8). In such cases, all student models underperform and collaboration becomes less effective. However, CoLLiS still demonstrates better performance over individual representations compared to prior LiDAR SemiSL approaches, as shown in Fig.8. This is because CoLLiS avoids over-reliance on a single knowledge source, reducing error reinforcement across students.
> ### Q3: Sensitivity to architectural or representational diversity
> When both representations and architectures are identical, RR naturally approaches 1 and distillation weights converge to 0.5, gracefully falling back to symmetric distillation. In this case, performance does not degrade, but the collaborative advantage is reduced. This is expected, as collaborative frameworks generally rely on complementary disparities in inputs and architectures to introduce diverse inductive biases that improve generalization[2].
>
> We conducted a relevant experiment in Tab. 11 (Appendix B.1), where we deploy two heterogeneous networks on the same representation. The results show that even without representational diversity, CoLLiS yields consistent gains over LaserMix on ScribbleKITTI (+2.5/+0.7 mIoU at 1%/10% labels), confirming that architectural diversity alone is sufficient to provide collaborative benefit. The gains are less pronounced than in the multi-representation setting. This motivates our default design choice of combining diverse representations, each of which captures complementary geometric cues beyond architectural diversity.
>
> Reference:
>
> [2]: A Good Student is Cooperative and Reliable: CNN-Transformer Collaborative Learning for Semantic Segmentation, Zhu et.al., ICCV'23
> ### W1: Minor typographical errors and dense mathmatical notations
> We appreciate the reviewer's feedback and will fix them in the revision.
> ### W2: Domain specificity of cross-representation mappings
> We acknowledge that the cross-representation mappings are LiDAR-specific by design. However, they are one instantiation of a more general consistency mechanism for training with multiple inputs with complementary disparities, which could be adapted to other multi-modal settings, such as radar-camera fusion. We see this as a promising direction for future work, as briefly discussed in Sec. 4.7.

---

> > ### Author Rebuttal · Reviewer_ZsAL · 2026-04-02
> >
> > The authors fully resolved my concerns, and I keep my original recommendation.

---

> > > ### Author Response · Authors · 2026-04-02
> > >
> > > We sincerely thank the reviewer for their valuable feedback. We are pleased that our responses have adequately addressed all of the reviewer’s concerns. We also greatly appreciate the reviewer’s engagement and positive assessment throughout the review process.

---

### Official Review · Reviewer_6rJV · 2026-03-13

**Soundness:** 3
**Presentation:** 3
**Significance:** 3
**Originality:** 2
**Overall Recommendation:** 4
**Confidence:** 5

**Summary:**

This paper examines semi-supervised LiDAR semantic segmentation and argues that existing two-step pseudo-labeling methods are prone to confirmation bias because they depend on a single pseudo-label source. To address this, the authors propose CoLLiS, a collaborative learning framework that jointly trains multiple LiDAR representations, such as range/frustum, polar, and voxel, in a unified one-step setting. The framework uses consensus-based augmentation, adaptive pseudo-labeling, and distillation to adjust supervision according to student agreement and reliability. Experiments on nuScenes, SemanticKITTI, and ScribbleKITTI across different label ratios show consistent improvements over prior LiDAR semi-supervised methods, especially in low-label settings. The paper also analyzes pseudo-label quality, confirmation bias, computational efficiency, scalability to additional representations, and compatibility with methods such as IT2.

**Compliance With Llm Reviewing Policy:**

Affirmed.

**Final Justification:**

My concerns are properly addressed. I would like to keep my rating.

**Key Questions For Authors:**

- If the authors can show low variance across multiple runs (with different label seeds), I would view the empirical claims as substantially stronger.
- Does the advantage of the proposed method remain under a fully supervised method?
- Why is the chosen confidence-count-based reliability measure preferable to alternatives such as entropy-based weighting?
- In addition to these questions, please refer to weaknesses.

**Limitations:**

yes

**Strengths And Weaknesses:**

Strengths
- The problem is important and the contribution is potentially significant. Reducing annotation cost for LiDAR segmentation is highly relevant for autonomous driving and 3D perception, where dense labeling remains extremely labor-intensive.
- The paper is experimentally strong and fairly convincing in terms of soundness. It evaluates the method on three standard benchmarks, multiple label ratios, and several additional settings.
- Overall presentation is also nice.

Weaknesses
- The originality is limited. The work is a combination of existing techniques (collaborative learning, confidence filtering, and adaptive augmentation) for multi-representation LiDAR SemiSL.
- Some core design choices are heuristic, which weakens the technical justification. The Absolute/Relative Reliability measures, thresholding rules, and adaptive weights are intuitive, but the paper does not fully explain why these particular formulations are the right ones.

---

> ### Author Rebuttal · Authors · 2026-03-30
>
> We sincerely thank the reviewer for the thoughtful and constructive evaluation. We especially appreciate the recognition of the importance of the problem, as well as the positive feedback on the strength of our empirical evaluation across multiple benchmarks and label regimes. We also thank the reviewer for the encouraging comments on the clarity and presentation of the paper. We address each concern below and will include all discussion and additional results in the revision.
> ### Q1: Low variance across multiple runs
> As requested by the reviewer, we ran CoLLiS with **3 random seeds** on two benchmarks with 1% labels. Results are reported below. The observed variance is small and consistent across different representations.
> ||F-Range|Polar|Voxel|
> |-|-|-|-|
> |nuScenes 1%|63.0(±0.5)|57.7(±0.7)|60.9(±0.4)|
> |SemKITTI 1%|55.8(±0.7)|46.8(±0.3)|53.1(±0.4)|
>
> ### Q2: Advantage under fully supervised setting
> Yes, Tab.9 in the paper shows CoLLiS consistently outperforms standalone baselines even under full supervision.
> ### Q3: Confidence-count vs. entropy-based weights
> Entropy-based weighting shares a similar limitation to mean-confidence measurement: both aggregate uncertainty across all classes, which can over-smooth the comparison signal between students of similar calibration. Moreover, maximum confidence can be sensitive to overconfidence spikes, particularly in semi-supervised settings with scarce labels. Empirically, we found that confidence-count-based reliability estimation leads to more stable and robust training.
> ### W1: Novetly beyond combination of existing techniques
> We respectfully clarify that the novelty of CoLLiS goes beyond simply combining existing components. The use of multiple LiDAR representations in the semi-supervised setting **remains underexplored**. The primary reason is the poor scalability: naïve extensions of two-step teacher–student methods to multiple representations introduce compounding pseudo-label conflicts and training instability. The key insight of CoLLiS is a **streamlined coequal-student paradigm** that eliminates the teacher–student hierarchy and **scales seamlessly to additional representations** for consistent performance gains, as demonstrated in Tab.4. However, naïve collaboration alone is insufficient in semi-supervised settings due to confirmation bias. Our reliability modeling provides a principled way to filter valid pseudo-labels under varying student strengths, while CDA adaptively self-regulates scene diversity to avoid under- and over-fitting. These components are **not independently** applicable off-the-shelf. Their **synergy** is what enables stable and effective multi-representation collaboration. The scalability of our framework makes it well-suited to grow alongside advances in LiDAR-based scene understanding. We further believe CoLLiS has **broader potential beyond LiDAR domain**, for instance, extending to radar representations or full multi-modal training, and we see this as a promising direction for furture work.
> ### W2: Rationale behind design choices
> We acknowledge the reviewer's concern and provide a theoretical grounding for our design choices. The framework stems from using **prediction confidence** as a proxy for uncertainty estimation. From a Bayesian perspective, an ideal measurement of reliability for a student model $s$ with parameters $\omega$ at input $x$ is the posterior probability of correctness $p(y_s(x)=y|x, \omega)$, where $y_s(x)$ denotes the prediction and $y$ the ground-truth label. In practice, this quantity is intractable. As neural networks are known to be imperfectly calibrated[0], their predicted uncertainty $\max_y p(y|x,\omega)$ does not accurately reflect the true probability of correctness. Consequently, the models may exhibit over- or underconfidence, particularly in semi-supervised settings. However, provided that models are trained with a non-trivial amount of labeled data, calibration errors predominantly affect the **magnitude** of confidence scores rather than their **relative ordering**[1]. Consequently, while absolute confidence values may be inaccurate, relative confidence comparisons across models remain informative: when one model consistently produces higher-confidence predictions than another on the same inputs, its predictions tend to be more trustworthy. This observation directly motivates our Relative Reliability measure, which compares confidence rankings between students rather than relying on absolute scores, and similarly justifies the adaptive weights for distillation. We acknowledge that this assumption may become weaker under severe distribution shift or extreme label scarcity, we therefore use the confidence regularization loss and Absolute Reliability (AR) to safeguard against such edge cases.
>
> *References*:
>
> [0]: Can you trust your model’s uncertainty? evaluating predictive uncertainty under dataset shift. Ovadia et al., Neurips'19
>
> [1]: On calibration of modern neural networks.Guo et al., ICML'17

---

> > ### Author Rebuttal · Reviewer_6rJV · 2026-04-03
> >
> > Thanks for the rebuttal.
> >
> > I would like to keep my rating as weak accept.

---

> > > ### Author Response · Authors · 2026-04-03
> > >
> > > We thank the reviewer for the constructive feedback and positive assessment of our work. We are pleased that all concerns have been fully resolved.

---

### Decision · Program_Chairs · 2026-04-30

**Decision:**

Accept (regular)

**Comment:**

The paper proposes a semi-supervised learning framework to improve LiDAR-based 3D semantic segmentation with limited annotated data. All the reviewers found that the paper is technically sound with substantial contributions to the community, and the rebuttal satisfactory addressed their concerns. The AC agrees with the reviewers’ consensus to accept this paper.